# Mode of Delivery and Neonatal Outcome in Adolescent Pregnancy (13–16 Years Old) Associated with Anemia

**DOI:** 10.3390/medicina58121796

**Published:** 2022-12-06

**Authors:** Ana Veronica Uzunov, Monica Mihaela Cîrstoiu, Diana Cristina Secară, Antoniu Crîngu-Ionescu, Alexandra Matei, Claudia Mehedințu, Valentin Nicolae Varlas

**Affiliations:** 1Doctoral School of “Carol Davila”, University of Medicine and Pharmacy, 4192910 Bucharest, Romania; 2Faculty of Medicine, “Carol Davila” University of Medicine and Pharmacy, 050451 Bucharest, Romania; 3Department of Obstetrics and Gynecology, University Emergency Hospital Bucharest, 050098 Bucharest, Romania; 4Department of Obstetrics and Gynecology, Saint Pantelimon Clinical Emergency Hospital, 021659 Bucharest, Romania; 5Department of Obstetrics and Gynecology, Filantropia Clinical Hospital, 011171 Bucharest, Romania

**Keywords:** childhood pregnancy, anemia, preterm birth, cesarean section, vaginal delivery, low birth weight

## Abstract

*Background:* Adolescent pregnancy represents an important public-health problem due to its maternal and fetal outcomes. Adolescent patients are predisposed to multiple obstetrical complications, including anemia and preterm birth which has a higher incidence among this population; withal, in the specialty literature, anemia is considered to be a risk factor for preterm delivery; furthermore, multiple studies have demonstrated that a very young age is an independent risk factor for preterm birth. *Objectives:* The study aims to reveal if anemia during adolescent pregnancy has a negative impact on the time and mode of delivery and newborns’ outcomes. *Patients and methods:* We performed a retrospective multicentric study on adolescent pregnancy. We analyzed 172 patients aged between 13 and 16 years who delivered in two large tertiary hospitals between 1 October 2018 and 15 April 2022. We divided the patients into two groups—a study group (n = 64) with anemia and a control group (n = 108) without anemia. We evaluated the modes of delivery, the times of birth, and the neonatal outcomes by 1-min newborn’s Apgar score, neonatal intensive-care unit (NICU) admission, and the newborns’ weights. *Results:* The rate of cesarean section was higher in patients with anemia than in the control group (45.31% vs. 38.88%, *p* < 0.001). We found that patients between 13 and 16 years diagnosed with anemia have a higher risk of preterm birth than those without anemia (35.93% vs. 21.29%, *p* < 0.001); however, an increased rate of LBW neonates was observed in the anemic adolescent group ≤14 years (*p* < 0.001). Regarding the newborns’ 1-min Apgar score, NICU admission, no statistically significant differences were recorded between the two groups according to the severity of anemia. In the anemic patients’ group, prenatal screening was identified in 9.37% of cases, while in the control group, in 16.67% (*p* = 0.034), which represents negative predictive factors, along with a low socio-economic status for the presence of anemia in young adolescent patients. *Conclusions:* Anemia is a risk factor for preterm birth, LBW, and cesarean section in young adolescent pregnancy. The association of lack of prenatal care and low socio-economic status worsens maternal and neonatal outcomes.

## 1. Introduction

The World Health Organization (WHO) defines adolescence as the life time between 10 and 19 years [1,2]; this represents a transition period from childhood to adulthood and includes biological, psychological, and social changes [3,4]. Adolescent pregnancy has a serious impact on maternal and neonatal status, but it also has side-effects on society, economics, and the next generations. According to WHO reports, in developing countries, every year, there are approximately 12 million adolescents between 15 and 19 years old and at least 777,000 girls under 15 years who give birth [5,6]. Adolescent pregnancy represents a serious public-health concern because the associated neonatal and maternal outcomes are negatively impacted. Newborns from adolescent patients have a higher risk of perinatal death, preterm delivery, low birth weight (LBW), and/or congenital anomalies. Regarding the obstetrical complications associated with adolescent pregnancy, maternal death, anemia, or preeclampsia are known to be more frequent [3,7,8]; in addition, since adolescents have biological and physiological immaturity, pregnancy increases their nutritional needs, so the anemia may be more accentuated [3].

As we talk about anemia, WHO defines it as the hemoglobin (Hb) concentration of less than 11 g/dL during pregnancy [3,9]. The geographical, ethnic, cultural, socio-economic, dietary, and methodological differences explain the large variation in the prevalence of anemia in pregnant teenagers in the various studies; thus, the overall prevalence of anemia in pregnant adolescent women ranges from 25% to 41.27%, with the highest values recorded in India at 70–76% [3,10,11,12,13]. Another study that analyzed 235 pregnant patients with anemia found that the highest incidence of anemia was in patients under 20 years, with a rate of 39.5% [14]; also, a research that included over 850,000 women aged under 25 years shows that the incidence of anemia is higher as the age decreases, such that patients aged 15 years or younger had an almost 40% increased risk of anemia compared with women between 20 and 24 years [10,14]. Anemia has a negative impact on pregnancy, as various studies reported that women diagnosed with anemia have a higher risk of preterm birth—that is, delivery before 37 completed weeks of gestation—than women without anemia [12,15,16,17,18,19,20,21,22,23,24,25]. Multiple risk factors have been described for preterm birth, but adolescent pregnancy represents an independent risk factor [11,26,27]. Studies have shown that anemia has an incidence of 76% in pregnant adolescents [28,29]; therefore, as early pregnancy represents an independent risk factor for preterm birth and there are assumptions that anemia also represents a risk factor for preterm birth, we may suppose that anemia in very young pregnant patients has an additional negative impact on neonatal outcomes.

Other studies show maternal anemia is associated with a higher risk of cesarean sections, low Apgar score, preterm birth, and/or preeclampsia [15,16,28,29,30,31]. Due to the higher incidence of adverse maternal and fetal outcomes caused by anemia in pregnant adolescent patients, it would be essential to have more conscious prenatal care for this population [32].

Although Romania is part of a group of developing/emerging countries with high-income economies, we have the highest adolescent pregnancy rate, ranking second in the European Union (EU). In the case of adolescents aged <14 years, the data on births are insufficient because, in recent years, the rate of these births has decreased at the EU level. In Romania, the prevalence rate of births among adolescent girls younger than 15 was 4%, with wide variability at the national level [33].

The most important key points of the study are to show whether the degree of anemia is a risk factor for preterm birth, if anemia affects the mode of delivery in very young pregnant patients, as well as neonatal outcomes in these patients.

## 2. Materials and Methods

We performed a multicentric, observational, retrospective study to evaluate the impact of anemia on young adolescents (13–16 years) and its implications on the mode of delivery and newborns’ outcomes. Anemia was recorded as a hemoglobin level under 11 g/dL, according to WHO. The hemoglobin level was established at admission to the hospital. We analyzed the modes of delivery (vaginal or cesarean section), the times of delivery, the neonatal status by 1-min Apgar score, the newborns’ weights, and the need for admission to the neonatal intensive-care unit (NICU); also, we assessed whether anemia has any impact on these.

### 2.1. Study Design

The subjects enrolled in the study were patients aged 13–16 years who delivered in two large tertiary hospitals in Bucharest between 1 October 2018 and 15 April 2022. The study included 172 patients whom we divided into two groups—a study group of 64 patients diagnosed with anemia and a control group of 108 patients without anemia.

Inclusion criteria into the study were: patients aged between 13–16 years old who delivered in our units and who signed themselves or their legal tutor (in the cases of patients under 16 years old) the informed consent about the study. An additional inclusion criterion for the study group was anemia. Exclusion criteria included the refusal of the patient or the legal tutor to sign the informed consent. Pregnancies at <24 weeks of gestational age, according to the legislation, as well as those born to teenage girls between the ages of 17 and 19, were excluded from the study because we chose to study the highly vulnerable 13–16 age groups and with the increased rate of maternal–fetal morbidity. The patient selection and division are presented in Figure 1.

The medical research ethics committees of the University Emergency Hospital (no. 56150/31.10.2018) and Saint Pantelimon Emergency Clinical Hospital (no. 13009/11.05.2020), respectively, approved the study protocol. The ethical standards of the declaration of Helsinki were followed.

### 2.2. Clinical Evaluation and Data Collection

Maternal age was recorded at the time of delivery according to the legal papers of the patient. Gestational age (GA) was established by the last menstrual period (if known) or according to the ultrasound evaluation and report; also, a clinical exam was performed to establish the GA. Expulsions of the fetuses after completed 24 weeks gestation or weighing over 500 g were recorded as preterm births, while before 24 weeks were considered abortions and were not included in this study. Preterm delivery was defined, according to WHO, as a newborn delivered before 37 completed weeks of gestation.

The information regarding pregnancy, anemia diagnosis, and delivery and neonatal outcomes data were retrieved from the hospitalization sheets and the Base Data System from the two hospitals.

### 2.3. Statistical Analysis

Data analysis was performed using JASP 0.16.2.0 Software. The Shapiro–Wilk test of normality was performed to assess the normal data distribution through the variables. The *t*-test was performed to established the significance of the differences between groups. The Pearson correlation test was performed to establish the impact of parameters of interest on different variables. Results with *p* < 0.05 were statistically significant.

## 3. Results

From the total 16,686 births in the studied period, 715 (4.28%) were among adolescents aged between 13 and 19 years. We analyzed 172 patients aged between 13 and 16 years who delivered in our units. There were 64 patients diagnosed with anemia, representing 37.21% of the total number of patients in the study. Regarding the mode of delivery for both groups, we found that 101 patients had a vaginal delivery, representing 58.72%, and 71 (41.28%) needed a cesarean section; thus, the cesarean-section rate among patients with anemia was 45.31% (n = 29), while the control group was 38.89% (n = 42). We analyzed the distribution of preterm birth and anemia in patients aged 13–16 years (Figure 1). The total number of preterm newborns was 46, representing 33.14% of the total newborns, with 29 cases (45.31%) in the study group. The case stratification was constructed according to maternal age, area of residence, BMI class, parity, GA, the status of prenatal care, birth weight, and NICU admissions (Table 1).

The only patient aged 13 years was diagnosed with anemia. The anemia rate reported by age group was 42.31% in ≤14-year-old patients, 42.11% in 15-year-olds; while 16-year-olds had the lowest incidence of anemia, 32.58% (Table 2).

We also analyzed the type of anemia. Most of the patients were diagnosed with iron-deficiency anemia, 89.06%. Megaloblastic anemia and β-thalassemia minor were diagnosed in 3.13% and 7.81% of cases, respectively.

Regarding preterm birth, 35.93% were in patients with anemia compared to 21.29% in patients without anemia (*p* < 0.001). A significant percentage of patients with anemia (45.31%) were delivered by cesarean section, compared to 38.88% in the control group (*p* < 0.001). The patient with severe anemia, with a hemoglobin level below 7 g/dL, had a vaginal delivery, and the gestational age was over 37 weeks (Table 3).

We determined the incidence of preterm birth and term delivery and the incidence of cesarean section and vaginal delivery in relation to patients’ ages (Table 4); therefore, the highest rate of preterm birth occurred in the ≤14 and 15 years group and the lowest rate in the 16 years group, which explains the increased vulnerability of the first group. Regarding the mode of delivery, most patients who needed a cesarean section were patients aged 16 years (51.72%), while patients aged 15 and ≤14 years had a cesarean section in 37.5% and 45.45% of cases, respectively. Vaginal delivery was more frequent, with a majority in the group of patients aged 15 years (62.5%).

Regarding the neonatal outcome, it was only one case of stillbirth in the study group. There were no statistically significant differences between the 1-min Apgar score in the newborns in the study group and those in the control group; withal, only 25.64% of newborns from the study group had a 1-min Apgar score of 10, while in the control group, the percentage was 74.36% (Table 5).

The neonatal condition was also evaluated in relation to the 1-min Apgar score and maternal age (Table 6). A reassuring Apgar score between 7 and 10 was given in 96.87% of the study group. An abnormal Apgar score was given in a single newborn from a 15-year-old patient. The only case of stillbirth was from a 16-year-old patient.

As we talk about neonatal status, the incidence of newborns weighing less than 2.500 g was higher in the anemia group (25%) compared with the control group (20.37%). Most newborns from the patients with anemia had a normal birth weight (75%), while LBW was observed in 21.87% of cases (Table 7).

The neonatal weight was also analyzed according to maternal age (Table 8); thus, most patients with normal-weight newborns came from the 16-year-old group (89.66%). Fetuses weighing less than 1500 g (ELBW and VLBW) came from 15-year-old patients. The rate of newborns with LBW was statistically significantly higher in the group of patients aged ≤14 years (36.36%), compared to 12.5% in patients aged 15 years and 10.34% in those aged 16 years (*p* < 0.001).

## 4. Discussion

Pregnancy among adolescents is a public-health problem due to the increased risk of adverse outcomes. In our study, we selected adolescents aged 13–16 because they represent a group with increased vulnerability, risk of complications during pregnancy and childbirth, inadequate prenatal medical care, negative social consequences, increased risk of school abandonment, and the impossibility of supporting their children.

Previous studies have shown that anemia and low iron stores are more prevalent in pregnant adolescents [3,34,35]. During pregnancy, the expansion of maternal blood and the requirements for iron in fetal tissues lead to an increased need for iron [11]. These lead to reduced fetal oxygenation and poor birth outcomes, such as prematurity, LBW, or stillbirth [11,36,37,38,39,40]. Pregnant adolescents are at higher risk of developing anemia due to the rapid growth and important biological changes of both the fetus and mother [41,42] or to improper dietary habits and lack of prenatal care [11].

The main reason for adverse neonatal and maternal outcomes may be poor socio-economic status, lack of prenatal care, or inappropriate lifestyle [11]. Casanueva et al. analyzed 163 adolescents aged between 11 and 17 years and showed that late prenatal care was associated with an increased risk of maternal anemia [43]. In our study, the lack of prenatal care was high, with a rate of 90.63% in the study group, while in the control group, the rate was 83.33%. Most of the adolescent patients aged between 13 and 16 who delivered in our units were patients from the Roma minority, and, as is known, their tradition is to get married and have children at a very young age.

According to the guidelines, mild anemia is defined as hemoglobin concentration between 9.0 g/dL and 10.9 g/dL, moderate anemia between 7.0 g/dL and 8.9 g/dL, and severe below 7.0 g/dL [44]. A study conducted in 12 clinical centers and 19 hospitals in the United States revealed that the risk of maternal anemia is increased for younger adolescents (age ≤ 15.9) [45]. In our study, the prevalence of anemia among patients aged 13–16 years was 37.2%, representing a significant incidence among these patients, with an observational peak among patients aged 16 years, where anemia had a percentage of 45.31% from all patients in the study group. Regarding these data, our patients with mild anemia were diagnosed in 78.12% of cases, with moderate anemia in 20.31%, and severe anemia in 1.56%. Pinho-Pompeo et al. highlighted the prevalence rates of mild, moderate, and severe anemia of 65.6%, 33.86%, and 0.52%, respectively [46]. Other studies revealed a high variability for moderate and severe forms (50.9% and 7.1% vs. 31.2% and 20.5%) [47,48]; furthermore, the study by Ganchimeg et al. showed an anemia rate for severe forms of 2.8% at <15 years of age vs. 2.2% at 16–17 years of age [2].

Studies showed heterogeneity in the geographical distribution regarding the type of anemia [5,48,49]; however, health programs for anemia in pregnant adolescents must identify the types of anemia and focus on correcting the deficiency of iron, folate, and micronutrients. Since 92.19% of the total number of anemic patients have either a form of iron deficiency or megaloblastic anemia, the early administration of iron, folic acid, and vitamin B12 supplements is the key to success in the management of these forms of anemia diagnosed in pregnant adolescents.

Multiple studies evaluated the mode of delivery among adolescent patients, but the results vary and may be contradictory depending on demographic characteristics [7,45,50,51]. Most reports demonstrate that cesarean section in developed countries is less frequent in patients under 18 years [7,45], while in developing countries the rate of cesarean section is higher in adult and adolescent patients, representing almost one-third of all surgeries [52,53]. A study analyzing 343 patients aged between 12 and 17 showed that half required cesarean section [50]. Another study found a higher cesarean rate at <15 years of age (28.1%) compared to 23% at 16–17 years [2]. In our study, the incidence of cesarean section was 41.28%. The distribution by age group showed a higher rate of cesarean in patients aged 16 (51.72%) and ≤14 (45.45%). The increased rate of cesarean sections among adolescents is explained by the fact that the younger the mother’s age, the greater the risk of labor dystocia, the existence of a modified bony pelvis, inadequate uterine response, and, sometimes due to the existence of the tutor, the mode of delivery can be influenced.

Anemia is an independent risk factor for cesarean section [15,16,25,31,54]. International consensus guidelines have recently defined cesarean section as surgery with moderate to high blood loss (>500 mL); therefore, it has a higher risk of blood transfusion compared to vaginal delivery [52,55,56]. In our study, vaginal birth was predominant (58.72%), but, as we analyzed the cesarean rate, we found that the incidence was higher in anemic patients (45.31%) compared to those without anemia (38.89%) (*p* < 0.001); therefore, our study’s high cesarean-section rate, especially in anemic patients, may be explained by the need for emergency surgery on fetal or maternal indications.

Anemia, especially in moderate and severe forms, is associated with worsening neonatal outcomes. It is known that anemia may increase the rate of preterm birth [12,15,16,19,20,21,22,23,24,25]. A large-scale study revealed a 10.8% rate of preterm birth in adolescent pregnant women, with the incidence increasing as age decreases in patients ≤15 years, at the rate of 14.6% [14]. In our study, preterm birth had an increased incidence in patients aged ≤14 and 15 (45.45% and 45.83%, respectively); hence, very young age represents an independent risk factor for preterm birth, and so does anemia among these patients.

Research that enrolled 36 pregnant adolescents with anemia and 86 without anemia showed that patients with anemia had a higher incidence of preterm birth (13.8%) than those from the control group [41]; despite this, we found a preterm birth rate of 26.74%, of which 35.93% occurred in anemic patients; furthermore, significant obstetrical conditions, including preeclampsia, preterm labor, or birth, are related to biological and physiological immaturity of adolescents, which consists of impaired vascular adaptation of the maternal spiral arteries during the endovascular trophoblast invasion [41,57]. We identified the association of preeclampsia (5.81%) and did not highlight an increased risk of puerperal endometritis and systemic infections.

Apgar score, NICU admission, and birth weight were analyzed to evaluate the neonatal status. Anemia has a significant negative impact on neonatal status; thus, newborns from anemic mothers are at a higher risk of low Apgar scores [41,49]. In contrast, another study that included 1.407 pregnant adolescents did not find any statistically significant impact of anemia on the Apgar score [32]; data also supported by our study.

Studies are controversial when discussing maternal age’s influence on the Apgar score. Ogawa et al. showed that newborns from adolescent patients have a higher rate of low Apgar scores than adult patients, while Yadav et al. found no difference [58,59]. Our study established that in patients aged 15 years, 8.32% of newborns had a 1-min Apgar score <7, which may be justified by the lack of biological and physiological development of the patients but may also be due to anemia.

Gestational anemia has a negative impact on fetal weight, evidenced by an increased rate of LBW [41,49,60]. Numerous studies have demonstrated a higher incidence of LBW and ELBW among adolescents than adults [15,61,62]. Jusoh et al. [63] found a 24% LBW rate in newborns of anemic adolescent mothers, while our study highlighted a rate of 21.87%; instead, Muñoz et al., looking at the impact of anemia on adolescent patients, found no correlation between low hemoglobin levels and the risk of LBW [55].

In addition, maternal age also significantly influences the weight of the newborn. An increase in the prevalence of preterm birth and severe neonatal conditions was observed with decreasing maternal age below 15 years; thus, the incidence of LBW in the age groups <15 years and 15–16 years was 14.6% vs. 12.4%, preterm birth 11.2% vs. 8.6%, and severe neonatal conditions (ELBW, VLBW, stillbirth, GA <30 weeks, and abnormal 1-min Apgar score) of 3.6% vs. 2.7% [2]. Research that analyzed 640 adolescents found that 1.1% of patients gave birth to a newborn under 1000 g [61], while in our work, ELBW was determined in only 5.26% of cases and only in patients aged 15 years. Considering all this, it is almost certain that young maternal age plus anemia can have a negative impact on infant weight; however, an increased rate of LBW neonates was observed in the anemic adolescent group ≤14 years (*p* < 0.001).

Prematurity was the most frequent cause of admission to the NICU of newborns of adolescent mothers; thus, the NICU admission rate varies between 13.3% and 16.36% [64,65], while in our study, the rate was 9.38% for newborns in the study group and 8.33% for those in the control group. A possible explanation is probably a higher rate of complicated pregnancies than our study.

This study has some limitations that should be considered as caveats. The sample size is small; therefore, randomized controlled trials would be impossible to conduct to investigate this type of issue because no intervention is conducted to randomize patients. In addition, the lack of information about pregnancy evolution, especially hemoglobin levels during pregnancy, represents another study limitation.

## 5. Conclusions

Anemia represents a risk factor for preterm delivery in patients between 13 and 16 years old; also, anemia in young adolescent pregnant patients has a negative impact on mode of delivery and neonatal outcome. As a consequence of these factors, we must strive to improve and implement new strategies to prevent adolescent pregnancies; also, it is important to have proper prenatal care plans to evaluate and treat very young patients. The implication for medical providers, families, and society is that this is an imperative need.

## Figures and Tables

**Figure 1 medicina-58-01796-f001:**
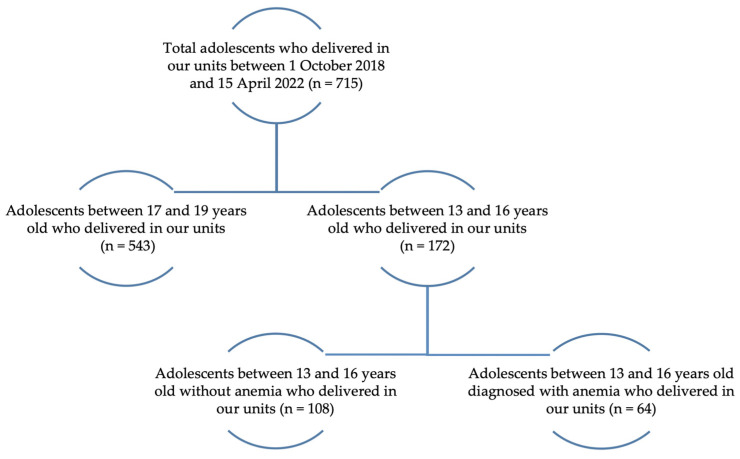
Flowchart of the distribution of the patients in the study.

**Table 1 medicina-58-01796-t001:** Description of study participant group structure and characteristics.

	Study Group (<11 g/dL) (n = 64)	Control Group (>11 g/dL) (n = 108)	*p*-Value *
Maternal age (mean ± SD)	15.26 ± 0.78	15.41 ± 0.72	<0.001
Area of residence (n, %)			
Urban	31 (48.43%)	38 (35.18%)	0.009
Rural	33 (51.57%)	70 (64.82%)	
Maternal weight (n, %)			
Normoponderal	63 (98.43%)	106 (98.14%)	0.045
Overweight (BMI—25–30 kg/m^2^)	1 (1.57%)	2 (1.86%)	
Type of anemia (n, %)			
Iron-deficiency anemia	57 (89.06%)	-	N/A
Megaloblastic anemia	2 (3.13%)	-	
β-thalassemia minor	5 (7.81%)	-	
Parity (n, %)			
Primiparous	55 (85.93%)	90 (83.33%)	<0.001
Multiparous	9 (14.07%)	18 (16.67%)	
Prenatal care (n, %)			
Yes	6 (9.37%)	18 (16.67%)	0.034
No	58 (90.63%)	90 (83.33%)	
Mode of delivery (n, %)			
Cesarean section	29 (45.31%)	42 (38.89%)	<0.001
Preeclampsia	6 (9.38%)	4 (3.70%)	
Negative induction of labor	8 (12.5%)	2 (1.85%)	
Cephalo-pelvic disproportion	4 (6.25%)	12 (11.11%)	
Dystocia presentation	2 (3.12%)	2 (1.85%)	
Fetal distress	7 (10.94%)	16 (14.81%)	
Uterine scar after cesarean section	2 (3.12%)	6 (5.56%)	
Vaginal delivery	35 (54.69%)	66 (61.11%)	0.411
GA at delivery (weeks) (mean ± SD)	36.73 ± 3.20	37.12 ± 2.47	0.539
**Neonatal outcomes**			
Preterm birth (n, %)	23 (35.93%)	23 (21.29%)	<0.001
Birth weight (grams) (mean ± SD)	2829.84 ± 620.40	2888.13 ± 599.24	0.396
1-min Apgar score (mean ± SD)	8.48 ± 1.39	8.62 ± 1.65	0.812
NICU admission, (n, %)	6 (9.38%)	9 (8.33%)	0.980

* *p* < 0.05 was considered statistically significant. NICU—Neonatal Intensive Care Unit; GA—gestational age.

**Table 2 medicina-58-01796-t002:** Distribution of patients in relation to hemoglobin level and age.

Patient’s Age			Hemoglobin Level (n/%)				
		Control Group (>11 g/dL)	Study Group (<11 g/dL)				*p*-Value
			Total	Mild Anemia (9–10.9 g/dL)	Moderate Anemia (7–8.9 g/dL)	Severe Anemia (<7 g/dL)	
≤14 years	26	15 (57.69%)	11 (42.31%)	11 (22.00%)	-	-	0.251
15 years	57	33 (57.89%)	24 (42.11%)	17 (34.00%)	7 (53.84%)	-	0.455
16 years	89	60 (67.41%)	29 (32.58%)	22 (44.00%)	6 (46.16%)	1 (100%)	<0.001

**Table 3 medicina-58-01796-t003:** Distribution of patients according to the time and mode of delivery.

			Hemoglobin Level (n/%)				
		Control Group (>11 g/dL)	Study Group (<11 g/dL)				*p*-Value
			Total	Mild Anemia (9–10.9 g/dL)	Moderate Anemia (7–8.9 g/dL)	Severe Anemia (<7 g/dL)	
**Time of delivery**							
Preterm birth	46	23 (21.29%)	23 (35.93%)	17 (34.00%)	6 (46.15.%)	-	<0.001
Term delivery	126	85 (78.71%)	41 (64.07%)	33 (66.00%)	7 (53.85%)	1 (100%)	0.037
**Mode of delivery**							
Cesarean section	71	42 (38.88%)	29 (45.31%)	25 (50%)	4 (30.76%)	-	<0.001
Vaginal delivery	101	66 (61.11%)	35 (54.69%)	25 (50%)	9 (69.24%)	1 (100%)	0.411

**Table 4 medicina-58-01796-t004:** Patients’ distribution in relation to time and mode of delivery according to maternal age.

Cases		Patient’s Age				
		Control Group	Study Group			*p*-Value
			≤14 Years (n = 11)	15 Years (n = 24)	16 Years (n = 29)	
**Time of delivery**						
Preterm birth	46	23 (34.25%)	5 (45.45%)	11 (45.83%)	7 (24.13%)	<0.001
Term delivery	126	85 (65.75%)	6 (54.55%)	13 (54.17%)	22 (75.87%)	0.014
**Mode of delivery**						
Cesarean section	71	42 (38.88%)	5 (45.45%)	9 (37.5%)	15 (51.72%)	0.363
Vaginal delivery	101	66 (61.12%)	6 (54.55%)	15 (62.5%)	14 (48.28%)	<0.001

**Table 5 medicina-58-01796-t005:** Distribution of patients according to 1-min newborn’s Apgar score and hemoglobin level.

1-min Apgar Score		Cases		Hemoglobin Level (n, %)				
			Control Group (>11 g/dL)	Study Group (>11 g/dL)				
				Total	Mild Anemia (9–10.9 g/dL)	Moderate Anemia (7–8.9 g/dL)	Severe Anemia (<7 g/dL)	*p*-Value
Extremely low	0–3	3	2 (1.85%)	1 (1.56%)	1 (2%)	-	-	0.420
Abnormal	4–6	5	4 (3.70%)	1 (1.56%)	1 (2%)	-	-	0.403
Reassuring	7–10	164	102 (94.44%)	62 (96.87%)	48 (96%)	13 (100%)	1 (100%)	0.638

**Table 6 medicina-58-01796-t006:** Newborns distribution in relation to 1-min Apgar score and maternal age.

1-min Apgar Score		Cases		Patient’s Age				
			Control Group (n = 108)	Study Group				*p*-Value
				Total	≤14 Years (n = 11)	15 Years (n = 24)	16 Years (n = 29)	
Extremely low	0–3	3	2 (1.85%)	1 (1.56%)	-	1 (4.16%)	-	0.667
Abnormal	4–6	5	4 (3.70%)	1 (1.56%)	-	1 (4.16%)	-	0.055
Reassuring	7–10	164	102 (94.44%)	62 (96.87%)	11 (100%)	22 (91.66%)	29 (100%)	0.055

**Table 7 medicina-58-01796-t007:** Distribution of patients according to newborns’ weights and hemoglobin levels.

Newborns’ Weights		Cases		Hemoglobin Levels (n, %)				
			Control Group (>11 g/dL)	Study Group				
				Total	Mild Anemia (9–10.9 g/dL)	Moderate Anemia (7–8.9 g/dL)	Severe Anemia (<7 g/dL)	*p*-Value
ELBW	<1.000 g	3	2 (1.85%)	1 (1.56%)	1 (2.00%)	-	-	0.003
VLBW	1.000–1.499 g	2	-	1 (1.56%)	1 (2.00%)	-	-	N/A
LBW	1.500–2.499 g	32	20 (18.51%)	14 (21.87%)	11 (22.00%)	3 (23.07%)	-	0.632
NBW	2.500–4.000 g	134	85 (78.70%)	48 (75.00%)	37 (74.00%)	10 (76.93%)	1 (100%)	0.867
	>4.000 g	1	1 (0.92%)	-	-	-	-	N/A

**Table 8 medicina-58-01796-t008:** Newborns’ distribution in relation to their weights and maternal ages.

Newborns’ Weights		Cases		Patients’ Ages (n/%)			
			Control Group	≤14 Years	15 Years	16 Years	*p*-Value
ELBW	<1.000 g	3	2 (1.85%)	-	1 (4.16%)	-	0.035
VLBW	1.000–1.499 g	2	-	-	2 (8.33%)	-	N/A
LBW	1.500–2.499 g	32	22 (20.37%)	4 (36.36%)	3 (12.50%)	3 (10.34%)	<0.001
NBW	2.500–4.000 g	134	83 (76.85%)	7 (63.63%)	18 (75.00%)	26 (89.66%)	<0.001
	>4000 g	1	1 (0.92%)	-	-	-	N/A

## Data Availability

The datasets used and analyzed during the current study are available from the corresponding author on reasonable request.

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
