# Peer review of "Mode of Delivery and Neonatal Outcome in Adolescent Pregnancy (13–16 Years Old) Associated with Anemia"

_medicina, 2022, doi:10.3390/medicina58121796_

Round 1

Reviewer 1 Report

Comments for the authors: In this paper, the authors investigated the mode of delivery and neonatal outcome
in childhood pregnancy, which is associated with anemia.
  Major concerns:  1, This study is well-designed, but the results are not surprising, so the novelty
of this paper is not significant. Numberless literature data can be found about
this topic, which proved the importance of anemia during pregnancy and birth
(e.g. Shoboo Rahmati , Milad Azami, Gholamreza Badfar , Naser Parizad,
Kourosh Sayehmiri. The relationship between maternal anemia during pregnancy
with preterm birth: a systematic review and meta-analysis.
2020 Aug;33(15):2679-2689. doi:10.1080/14767058.2018.1555811. Epub 2019 Apr 9.
Qiaoyi Zhang, Cande V Ananth, Zhu Li, John C Smulian.Maternal anaemia and
preterm birth: a prospective cohort study. International Journal of Epidemiology,
Volume 38, Issue 5, October 2009, Pages 1380–1389, https://doi.org/10.1093/ije/
dyp243. etc.).
2, The official number of the ethics license is missing. Minor remarks: - It would be more clear to represent the results on column charts. - I think that no conclusion can be drawn between the mode of delivery
(cesarean section or vaginal delivery) and anemia. Many other non-health
conditions also affect the mode of delivery.
- The style of the references is not uniform. (Sometimes it also includes
doi, e.g. 53, 61.) References 6, 57 are unidentifiable.

Author Response

Reply to Reviewer

Major concerns:  

1, This study is well-designed, but the results are not surprising, so the novelty of this paper is not significant. Numberless literature data can be found about this topic, which proved the importance of anemia during pregnancy and birth(e.g. Shoboo Rahmati , Milad Azami, Gholamreza Badfar , Naser Parizad, Kourosh Sayehmiri. The relationship between maternal anemia during pregnancy with preterm birth: a systematic review and meta-analysis. 2020 Aug;33(15):2679-2689. doi:10.1080/14767058.2018.1555811. Epub 2019 Apr 9. Qiaoyi Zhang, Cande V Ananth, Zhu Li, John C Smulian. Maternal anaemia and preterm birth: a prospective cohort study. International Journal of Epidemiology, Volume 38, Issue 5, October 2009, Pages 1380–1389, https://doi.org/10.1093/ije/dyp243. etc.). 

Answer: Thank you for your mention.

Adolescent pregnancy is a public health problem due to the increased risk of adverse outcomes. In our study, we selected adolescents aged 13-16 years because they represent a group with increased vulnerability, risk of complications during pregnancy and childbirth, inadequate prenatal medical care, negative social consequences, increased risk of dropping out of school, and inability to support their children. Data from the literature are extremely scarce regarding pregnancy in teenage girls under the age of 16.

2, The official number of the ethics license is missing. 

Answer: Thank you for your valuable remark; we introduced the ethical approval number. (please see the attached manuscript) Lines 125-127.

Minor remarks: 

- It would be more clear to represent the results on column charts. 

Answer: Thank you for your mention; We considered that the tables are the most appropriate to clearly highlight information specific to the studied groups.

.

- I think that no conclusion can be drawn between the mode of delivery (cesarean section or vaginal delivery) and anemia. Many other non-health conditions also affect the mode of delivery. 

Answer: Thank you for your comment; In our study, we found a higher cesarean section rate among anemic patients (45.31%) compared to those without anemia (38.89%) (p<0.001). Indeed, there are other factors that can influence the mode of delivery, given that in this study, we focused on the influence of anemia as the main factor.

- The style of the references is not uniform. (Sometimes it also includes doi, e.g. 53, 61.) References 6, 57 are unidentifiable.

Answer: Thank you for your remark; we corrected the structure of references according to your recommendations. (please see the attached manuscript).

Kindest regards

Authors

Reviewer 2 Report

I reviewed this paper with particular interest, bacause of I work with female adolescence and I think it is imortant to underline attention on childhood pregnancies and their potential complications both for the young mother and the baby. I think it is suitable for publication even if it is auspicable to carry on investigations in this specific field and collect more data in the future. 

Author Response

Reply to Reviewer

I reviewed this paper with particular interest, bacause of I work with female adolescence and I think it is imortant to underline attention on childhood pregnancies and their potential complications both for the young mother and the baby. I think it is suitable for publication even if it is auspicable to carry on investigations in this specific field and collect more data in the future. 

Thank you for your appreciation. Our study joins other studies regarding the maternal and neonatal outcomes in pregnant adolescents with anemia by referring to the age group 13-16 years which is the most vulnerable and in which the data in the literature are not completely systematized (especially under 16 years).

Kindest regards

Authors

Reviewer 3 Report

This paper appears to be an attempt to investigate whether neonatal outcomes are worse in adolescent pregnancies complicated by anaemia than in adolescent pregnancies not complicated by anaemia.  It seems likely to me that these risks would be combined and, therefore, increased, although I am not familiar enough with the relevant literature.  The authors do not make clear in the introduction about how well previous studies have investigated this.  Reference 19 appears to have addressed and answered this research question.  If this is correct, it would be useful if the authors could explain their rationale for repeating this study?

The aims and objectives of the study need to be clarified.  The objectives stated in the abstract appear to be very different to the aims stated in Line 76. 

I am not sure how table 6 onwards and the associated discussion relate to the stated objectives of the study as they do not make any comparisons in relation to whether or not there was an anaemia.

The only outcomes studied appear to be: mode of delivery, gestation, 1 minute Apgar score, and Birth weight.  The authors state (line 87) that the need for NNU admission was also assessed, but no data relating to that outcome are presented.

Methodology:

Were there any excluded patients?

How did they select the age group 13 to 16?  Why not include all adolescents (to age 19)?

Inclusion criteria are confused – Line 101 states that anaemia was an inclusion criteria – this doesn’t make sense as 108 weren’t anaemic.

Live births before 24 weeks were not included – why not?

It appears that still births and other pregnancy losses were not included – why?

There does appear to be 1 stillbirth in the data (Table 4) – This is confusing.  How were patients selected?

Results:

The main criticism of the paper is the lack of any robust or meaningful statistical analysis of the data.  Although the authors have stated that there were some differences between the anaemic and non-anaemic groups, they have not conducted or presented any statistical analysis to show that these differences are unlikely to be anything other than chance findings. 

For example, Caesarean section was performed in 45.31% of the anaemic group v 38.89% in the non-anaemic group.  No significance test or p value is given in the text (line 125). A p value of <0.001 is shown in table 1, but the type of significance test used is not explained.

There is a similar lack of significance testing for preterm birth. No p value appears in the text or table.  The difference in mean weight is very small and unlikely to be of any statistical significance. Birth Weight should be presented as a non-parametric variable (Median + range or IQR) it is never normally distributed. 

“Preterm birth” <37 weeks is s very crude definition of prematurity.  Can the authors describe the distribution of gestations between the 2 groups and do a Mann-Whitney test to investigate whether these are different? Gestation should be presented as a non-parametric variable (Median + range or IQR) it is never normally distributed.

Line 132 onwards describes an analysis performed using data that has not been referred to previously.  This should be included in the Methods section.

Table 1 – what definition of Obese is used?

Did the rows without p values reported show no statistical significance?  If so, what test was used?

Table 2 – It is very difficult to understand what this table is supposed to show.  There is no significance testing presented.  Any differences could easily be explained by chance.

The authors have attempted, in table 2, to investigate whether [Hb]  is associated with mode of delivery by dividing the groups into 7 different [Hb] bands.  No statistical analysis has been performed, so no meaningful conclusions can be reached about these data.  It would be better to use a Mann- Whitney test to look at the distribution of [Hb] in the babies born by caesarean section compared to those born vaginally.

Table 3 also has no significance testing presented.  The differences are very small and could be explained as chance findings.

Figure 2 is a repeat of the data presented in table 1.

Table 4 – This is also confusing.  For example, the authors have shown what percentage of babies with a 1 minute Apgar score of 10 were in each [Hb] band.  It would be much more interesting to show what percentage of each [Hb] band had which Apgar score.

The authors have attempted, in table 4, to investigate whether [Hb]  is associated with Apgar score by dividing the groups into 7 different [Hb] bands.  No statistical analysis has been performed.  It would be better to use some sort of correlation or regression analysis to explore this.

The authros report that the proportion of birth weight under 2,500g was higher in the anaemic pregnancies than the non-anaemic pregnancies (25% v 20.37%) – no significance test appears to have been performed to test whether this is likely to be a chance finding or not.

Tables 5 and 6 – similar comments to the other tables.

The authors try to explore whether there is a relationship between maternal age and mode of delivery – they should perform some sort of statistical analysis on this – eg – Mann Whitney.

The paragraph starting in Line 190 is very difficult to understand.  I am not clear what the authors are trying to describe and whether or not the differences they are trying to highlight are real and statistically significant, or just chance findings.  If they are interested in a possible association between 1 minute Apgar score and maternal age, then this could be easily assessed using simple correlation or regression analysis.

I make very similar criticisms of the attempt to investigate the relationship between maternal age and Birth Weight.

Discussion:

The Discussion is flawed because some of the conclusions reached by the authors are not supported by the study because of inadequate statistical exploration of their data.  They have made claims about differences that they have not shown to have occurred by chance.

The authors describe high rates of caesarean section in studies of adolescent birth in the developing world.  These are likely to be selective samples of the total population in those studies if they are hospital based as there is likely to be a significant number of home deliveries in those populations.

Study Limitations:

Line 363 says “randomized controlled trials with a large sample size should be performed with the same pattern to generate more accurate results”.  Randomised controlled trials would be impossible to perform to investigate this sort of issue as there is no intervention being made to randomise patients to.

Conclusions:

These cannot be justified on the basis of the lack of significance testing in this study.

Minor comments:

ABSTRACT

The authors state that they… “the moment of birth,” – what does that mean? (Line 29)

No p values are given in the abstract to support the associations that the authors believe that they have seen.

Line 30 and line 35 are a duplication of the same point.

INTRODUCTION

The authors state in line 49/50 “Newborns from adolescent patients have a higher risk of perinatal death, preterm delivery, low birth weight, or congenital anomalies.” This statement requires a reference to support it.

Line 59 “Also, a research that included…” should be “Also, a study that included…”

Line 59 “850.000 women” should this be “850,000” (There are several points in the manuscript when a comma is used instead of a full stop and vice versa).

The authors state in line 71 “Studies have shown that anemia has an incidence of 76% in pregnant adolescents [29,31].” They provide a different rate of anaemia in pregnant adolescents stated earlier in the paper (Line 58) “highest incidence of anemia was in patients under 20 years, with a rate of 39.5% [12].” The differences between these 2 reports should be discussed and explained.

MATERIALS AND METHODS

Line 88 “assayed if anaemia” should be “assessed whether anaemia”

Author Response

Reply to Reviewer

This paper appears to be an attempt to investigate whether neonatal outcomes are worse in adolescent pregnancies complicated by anaemia than in adolescent pregnancies not complicated by anaemia.  It seems likely to me that these risks would be combined and, therefore, increased, although I am not familiar enough with the relevant literature.  The authors do not make clear in the introduction about how well previous studies have investigated this.  Reference 19 appears to have addressed and answered this research question.  If this is correct, it would be useful if the authors could explain their rationale for repeating this study?

Answer: Thank you for your mentionReference 19 refers to adolescents (10-19 years), our study analyzed adolescents aged 13-16 years, because our country has a high incidence in pregnant adolescents and therefore, its impact is very important. Also, regarding the exact prevalence of anemia and pregnancy in patients aged under 15 years, there are inconstant data in low- and middle-income countries, due to underreported cases.

The aims and objectives of the study need to be clarified.  The objectives stated in the abstract appear to be very different to the aims stated in Line 76.  

Answer: Thank you for your mention; we corrected the problem.

I am not sure how table 6 onwards and the associated discussion relate to the stated objectives of the study as they do not make any comparisons in relation to whether or not there was an anaemia.

The only outcomes studied appear to be: mode of delivery, gestation, 1-minute Apgar score, and Birth weight.  The authors state (line 87) that the need for NNU admission was also assessed, but no data relating to that outcome are presented.

Answer: Thank you for your remark; we added the data regarding the admission of newborns to the NICU. (please see the attached manuscript) table 1 and lines 308-312.

Methodology:

Were there any excluded patients?

Answer: Thank you for your remark; there were not any excluded patients.

How did they select the age group 13 to 16?  Why not include all adolescents (to age 19)?

Answer: Thank you for your comment; we selected adolescents aged 13 to 16 years because they represent a vulnerable group that has significant problems in our country and the complications associated with pregnancy (lines 113-115).

Inclusion criteria are confused – Line 101 states that anaemia was an inclusion criteria – this doesn’t make sense as 108 weren’t anaemic.

Answer: Thank you for your mention; Our study was structured on two groups: a study group consisting of pregnant teenagers with anemia and a control group.

Live births before 24 weeks were not included – why not?

Answer: Thank you for your comment; live births before 24 weeks are considered abortions, therefore, they were not included (lines 112-113).

It appears that stillbirths and other pregnancy losses were not included – why? 

Answer: Thank you for your comment; pregnancy losses were not included because we studied only births. Stillbirths were included, and it was only one case.

There does appear to be 1 stillbirth in the data (Table 4) – This is confusing.  How were patients selected?

Answer: Thank you for your mention; Patients selected were all adolescents aged 13-16 years who gave birth in our centers, even stillbirths. There was only one case of stillbirth.

Results:

The main criticism of the paper is the lack of any robust or meaningful statistical analysis of the data.  Although the authors have stated that there were some differences between the anaemic and non-anaemic groups, they have not conducted or presented any statistical analysis to show that these differences are unlikely to be anything other than chance findings.  

For example, Caesarean section was performed in 45.31% of the anaemic group v 38.89% in the non-anaemic group.  No significance test or p value is given in the text (line 125). A p value of <0.001 is shown in table 1, but the type of significance test used is not explained.

There is a similar lack of significance testing for preterm birth. No p value appears in the text or table.  The difference in mean weight is very small and unlikely to be of any statistical significance. Birth Weight should be presented as a non-parametric variable (Median + range or IQR) it is never normally distributed.  

“Preterm birth” <37 weeks is s very crude definition of prematurity.  Can the authors describe the distribution of gestations between the 2 groups and do a Mann-Whitney test to investigate whether these are different? Gestation should be presented as a non-parametric variable (Median + range or IQR) it is never normally distributed.

Line 132 onwards describes an analysis performed using data that has not been referred to previously.  This should be included in the Methods section.

Answer: Thank you for your mention; we included it in the Methods section.

Table 1 – what definition of Obese is used?

Answer: Thank you for your remark; we used Quetelet’s index (BMI) to classify obese pregnant women At admission to the hospital for IOL, weight (kg) was measured with a calibrated weight scale, and height (m) with a calibrated stadiometer. Body mass index was calculated as weight (kg)/height squared (m2).

Did the rows without p values reported show no statistical significance?  If so, what test was used?

Table 2 – It is very difficult to understand what this table is supposed to show.  There is no significance testing presented.  Any differences could easily be explained by chance.

The authors have attempted, in table 2, to investigate whether [Hb]  is associated with mode of delivery by dividing the groups into 7 different [Hb] bands.  No statistical analysis has been performed, so no meaningful conclusions can be reached about these data.  It would be better to use a Mann- Whitney test to look at the distribution of [Hb] in the babies born by caesarean section compared to those born vaginally. 

Table 3 also has no significance testing presented.  The differences are very small and could be explained as chance findings.

Answer: Thank you for your mention; we entered the statistically processed data.

Figure 2 is a repeat of the data presented in table 1.

Answer: Thank you for your valuable remark; we removed figure 2.

Table 4 – This is also confusing.  For example, the authors have shown what percentage of babies with a 1 minute Apgar score of 10 were in each [Hb] band.  It would be much more interesting to show what percentage of each [Hb] band had which Apgar score.

The authors have attempted, in table 4, to investigate whether [Hb]  is associated with Apgar score by dividing the groups into 7 different [Hb] bands.  No statistical analysis has been performed.  It would be better to use some sort of correlation or regression analysis to explore this. 

The authros report that the proportion of birth weight under 2,500g was higher in the anaemic pregnancies than the non-anaemic pregnancies (25% v 20.37%) – no significance test appears to have been performed to test whether this is likely to be a chance finding or not.

Tables 5 and 6 – similar comments to the other tables.

The authors try to explore whether there is a relationship between maternal age and mode of delivery – they should perform some sort of statistical analysis on this – eg – Mann Whitney.

The paragraph starting in Line 190 is very difficult to understand.  I am not clear what the authors are trying to describe and whether or not the differences they are trying to highlight are real and statistically significant, or just chance findings.  If they are interested in a possible association between 1 minute Apgar score and maternal age, then this could be easily assessed using simple correlation or regression analysis.

I make very similar criticisms of the attempt to investigate the relationship between maternal age and Birth Weight.

Answer: Thank you for your recommendations; we did the statistic tests for all the tables. Data analysis was performed using JASP 0.16.2.0 Software. The results chapter was restructured, and the statistical analysis of the results was introduced.

Discussion:

The Discussion is flawed because some of the conclusions reached by the authors are not supported by the study because of inadequate statistical exploration of their data.  They have made claims about differences that they have not shown to have occurred by chance.

The authors describe high rates of cesarean section in studies of adolescent birth in the developing world.  These are likely to be selective samples of the total population in those studies if they are hospital-based as there is likely to be a significant number of home deliveries in those populations.

Answer: Thank you for your recommendations; according to your suggestions, we rewrote the discussion chapter more clearly. (please see the attached manuscript) Lines 201-317.

Study Limitations:

Line 363 says, “randomized controlled trials with a large sample size should be performed with the same pattern to generate more accurate results”.  Randomized controlled trials would be impossible to perform to investigate this sort of issue as there is no intervention being made to randomize patients to.

 Answer: Thank you for your recommendation; we made the correction. (please see the attached manuscript) Lines 314-315.

Conclusions:

These cannot be justified on the basis of the lack of significance testing in this study.

Answer: Thank you for your mention; the conclusions are based on statistically significant results after the introduction of the statistical analysis.

Minor comments:

ABSTRACT

The authors state that they… “the moment of birth,” – what does that mean? (Line 29)

Answer: Thank you for your mention; instead of “the moment of birth,” we introduced a time of delivery.

No p values are given in the abstract to support the associations that the authors believe that they have seen.

Answer: Thank you for your mention; we inserted the p-values

Line 30 and line 35 are a duplication of the same point.

Answer: Thank you for your mention; we corrected the problem.

INTRODUCTION

The authors state in line 49/50 “Newborns from adolescent patients have a higher risk of perinatal death, preterm delivery, low birth weight, or congenital anomalies.” This statement requires a reference to support it.

Line 59 “Also, a research that included…” should be “Also, a study that included…”

Line 59 “850.000 women” should this be “850,000” (There are several points in the manuscript when a comma is used instead of a full stop and vice versa).

The authors state in line 71 “Studies have shown that anemia has an incidence of 76% in pregnant adolescents [29,31].” They provide a different rate of anaemia in pregnant adolescents stated earlier in the paper (Line 58) “highest incidence of anemia was in patients under 20 years, with a rate of 39.5% [12].” The differences between these 2 reports should be discussed and explained.

Answer: Thank you for your mention; we solved the issues.

MATERIALS AND METHODS

Line 88 “assayed if anaemia” should be “assessed whether anaemia”

Answer: Thank you for your suggestion; we corrected the problem.

Kindest regards

Authors